# Effects of a Novel Nutraceutical Combination (Aquilea Colesterol^®^) on the Lipid Profile and Inflammatory Biomarkers: A Randomized Control Trial

**DOI:** 10.3390/nu11050949

**Published:** 2019-04-26

**Authors:** Mònica Domenech, Rosa Casas, Ana Maria Ruiz-León, Javier Sobrino, Emilio Ros, Ramon Estruch

**Affiliations:** 1Department of Internal Medicine, Institut d’Investigacions Biomèdiques August Pi Sunyer (IDIBAPS), Hospital Clinic, University of Barcelona, 08036 Barcelona, Spain; mdomen@clinic.cat (M.D.); rcasas1@clinic.cat (R.C.); amruiz@clinic.cat (A.M.R.-L.); 2CIBER Fisiopatología de la Obesidad y Nutrición (CIBEROBN), Instituto de Salud Carlos III (ISCIII), 28029 Madrid, Spain; eros@clinic.ub.es; 3Mediterranean Diet Foundation, 08021 Barcelona, Spain; 4Department of Internal Medicine, Fundació Hospital Esperit Sant, University of Barcelona, 08923 Barcelona, Spain; jsobrino@fhes.cat; 5Lipid Clinic, Endocrinology and Nutrition Service, IDIBAPS, Hospital Clinic, University of Barcelona, 08036 Barcelona, Spain

**Keywords:** nutraceuticals, hypercholesterolemia, C-reactive protein, red yeast rice, phytosterols, hydroxytyrosol, vitamin E, Aquilea Colesterol

## Abstract

Background: Cholesterol-lowering nutraceuticals are useful in the management of moderate hypercholesterolemia. Methods: In a parallel-group, randomized, placebo-controlled double-blind trial we evaluated the effects on plasma total cholesterol, low-density lipoprotein cholesterol (LDL-c), and inflammatory biomarkers of a nutraceutical combination (Aquilea Colesterol^®^) containing phytosterols (1.5 g), red yeast rice providing monacolin K (10 mg), hydroxytyrosol (5 mg), and plasma cholesterol values >5.17 mmol/L (>200 mg/dL) and LDL-c >2.97 mmol/L (>115 mg/dL). At baseline and at one and three months we recorded dietary habits; anthropometric parameters; blood pressure; lipid profile; fasting glucose; liver, renal, and muscle function tests, C-reactive protein (hs-CRP); and interleukin-6. Results: 13 men and 27 women (mean age 61.8 years) completed the trial; 20 participants received the nutraceutical and 20 received placebo. No adverse effects were noted. Compared to placebo, at one and three months the nutraceutical reduced total cholesterol by 11.4% and 14.1%, LDL-c by 19.8% and 19.7%, and apolipoprotein B by 12.4% and 13.5%, respectively (*p* < 0.001; all). hs-CRP decreased significantly (*p* = 0.021) in the nutraceutical group. Conclusion: The nutraceutical Aquilea Colesterol^®^ is useful for reducing total cholesterol, LDL-c, and inflammation in individuals with moderate hypercholesterolemia.

## 1. Introduction

To date, cardiovascular disease (CVD) continues to be the leading cause of morbidity and mortality worldwide [1]. There is compelling evidence that elevated total cholesterol and low-density lipoprotein cholesterol (LDL-c) are causally related to CVD [2], signaling hypercholesterolemia as an important modifiable cardiovascular risk factor [3,4]. Statins, the standard treatment for elevated blood cholesterol, have a large record of efficacy and safety [5], but the meta-analysis of Silverman et al. [5] indicates that LDL-c reduction translates into cardiovascular benefit independently of the intervention used [6]. Targets in the management of hypercholesterolemia depend on age, sex, previous history of CVD, and presence or absence of other cardiovascular risk factors [3,4]. As described in recent comprehensive reviews (7–9) and position papers (10–12), the lipid-lowering effect of most nutraceuticals may be useful in the management of individuals at low-to-moderate cardiovascular risk who maintain plasma cholesterol levels ≥5.17 mmol/L (200 mg/dL) and/or LDL-c ≥2.97 mmol/L (115 mg/dL) despite a healthy diet and overall lifestyle and do not qualify for statin therapy due to their low overall risk. The European Society of Atherosclerosis and the European Society of Cardiology guidelines for the management of dyslipidemia state that supplementation with cholesterol-lowering functional foods or nutraceuticals might be indicated in this situation [3]. These compounds are also appropriate alternatives for individuals with intolerance to statins or those who are not willing to take them [7]. 

The cholesterol-lowering nutraceuticals are compounds derived from foods or natural molecules that target metabolic pathways leading to a reduction in body cholesterol pools and upregulation of LDL receptor expression and are commercialized in pills, powder, or other medical forms and sometimes are consumed in fortified foods. As reviewed [7,8,9,10,11,12], the lipid-lowering effect of most nutraceuticals occurs via multiple mechanisms, and some of them disclose added pleiotropic effects to reduce cardiovascular risk, such as improved endothelial function, reduction of arterial stiffness, and induction of anti-inflammatory and anti-oxidant actions.

Among the nutraceuticals with demonstrated effects on hyperlipidemia, the most commonly utilized are plant sterols or phytosterols, red yeast rice (RYR), omega-3 fatty acids, soluble fibers, berberine, and policosanol, although only the first four compounds have been recognized by the Food and Drug Administration (FDA) and the European Food Safety Authority (EFSA) as safe and effective in cholesterol lowering [13]. Nutraceutical combinations are increasingly used in clinical practice and often they incorporate antioxidants or other bioactive molecules to obtain a putative benefit beyond cholesterol reduction. As an example, a marketed combination of berberine, policosanol, and RYR with the antioxidant carotenoid astaxanthin and coenzyme Q10 has shown its efficacy in improving the lipid profile in a meta-analysis of 11 randomized clinical trials including 3924 participants [14].

Concerning antioxidants that might be useful in nutraceutical combinations, the EFSA panel also considered that 5 mg of hydroxytyrosol and related molecules (oleuropein and tyrosol) from olive oil could contribute to normalizing high-density lipoprotein cholesterol (HDL-c) concentrations and protect LDL against oxidation [13]. In addition, there is increasing evidence on the role of vitamin E in overall health derived from the reduction of oxidative stress and neuroprotection [15]. Few studies have evaluated the effects on lipids and inflammation of these compounds in novel nutraceutical preparations. In the present study, we assessed the effects of a nutraceutical combining phytosterols, RYR, hydroxytyrosol, and vitamin E (Aquilea Colesterol^®^) on the lipid profile and inflammatory biomarkers (high-sensitive C-reactive protein (hs-CRP) and interleukin (IL)-6) in patients with hypercholesterolemia at low-to-moderate cardiovascular risk.

## 2. Materials and Methods

### 2.1. Study Design, Participants, Setting, and Intervention

This study was designed as a randomized, parallel-group, double-blind, placebo-controlled trial aimed to assess the efficacy of a novel nutraceutical combination (Aquilea Colesterol^®^) that contains 1.5 g phytosterols, 10 mg monacolin K from RYR, 5 mg hydroxytyrosol from olive leaf extract (*Olea europaea*), and 12 mg vitamin E in the reduction of total cholesterol and LDL-c in patients with primary hypercholesterolemia at low-to-moderate cardiovascular risk. The study protocol complied with the Declaration of Helsinki, was approved by the Institutional Review Board of Hospital Clinic (Barcelona, Spain), and was registered in the International Standard Randomized Controlled Trial Number (ISRCTN) register with the number 15149502 (www.controlled-trials.com). All participants provided written informed consent. 

Participants were men and women aged between 35 and 75 years at low (<5%) or moderate (5–9%) cardiovascular risk according to the Registre Gironí del Cor (REGICOR) risk score [16], with total plasma cholesterol between 5.17 and 6.47 mmol/L (200–250 mg/dL) and LDL-c >2.97 mmol/L (115 mg/dL) in the absence of lipid-lowering treatment. Exclusion criteria were prior intake of lipid-lowering drugs (statins, fibrates, resins, or ezetimibe) and/or any other product with possible lipid-lowering effects, such as RYR or plant sterols, in the last three months, and history of CVD defined by prior coronary heart disease (myocardial infarction, angina), stroke, or peripheral artery disease. Individuals with a clinical indication for lipid-lowering drugs and those at high (10–14%) or very high coronary risk (≥15%) according to the REGICOR score [16] were also excluded from the study.

Participants were recruited at the outpatient clinic of the Department of Internal Medicine, Hospital Clinic, Barcelona (Spain). We assessed for eligibility individuals consecutively attending the clinic between May and October 2018 and scheduled them for a screening visit by a physician to confirm inclusion/exclusion criteria and obtain signed informed consent. Participants underwent a two-week run-in period with instructions to follow a healthy diet to be maintained throughout the study. At the baseline visit, participants were randomized 1:1 into the intervention group, wherein they received a daily liquid stick of Aquilea Colesterol^®^ (see above), or the placebo group, according to a computer-generated random number sequence table. Both the nutraceutical combination and the placebo were taken daily before the main meal during 90 days (three months). All participants were scheduled for follow-up visits at one and three months.

### 2.2. Assessment of Adverse Effects

At the one- and three-month visits, the investigators assessed any adverse effects from the intervention according to a checklist of symptoms and, if necessary, gave advice on how to remedy them. The checklist included bloating, fullness, or indigestion as well as altered bowel habit or any other intervention-related symptom.

### 2.3. Procedures and Cardiovascular Risk Assessment

At baseline and at one and three months, we collected data on the medical history, medication use, dietary habits, and physical activity; measured anthropometric variables and office blood pressure (BP); and obtained fasting blood samples for the analyses of lipids and inflammatory biomarkers. Smoking status was categorized as never, current, or past smoking according to self-reports. Dietary habits were assessed according to adherence to the Mediterranean diet evaluated using a validated 14-point questionnaire [17]. Physical activity was assessed using a validated short version of the Minnesota questionnaire [18]. Height, weight, and waist circumference were measured with standard methods. Office BP was measured in triplicate with a validated semiautomatic oscillometer (Omron HEM-705CP; Hoofddorp, The Netherlands) following current guidelines [19]. Fasting blood samples were obtained by venipuncture at baseline and at 4 and 12 weeks, and aliquots of plasma and serum were stored at −80 °C until the analysis of inflammatory biomarkers. Glycemic control and liver, renal, and muscle function tests were determined by standard enzymatic techniques. Total plasma cholesterol and triglyceride concentrations were measured by enzymatic procedures and HDL-c was determined after precipitation with phosphotungstic acid and magnesium chloride in a clinical chemistry analyzer Advia 2400 (Siemens, Tarrytown, NY, USA). Plasma LDL-c concentrations were calculated using the Friedewald formula: LDL-c = total cholesterol − HDL-c − (triglycerides/2.21 for mmol/L or triglycerides/5 for mg/dL) [20]. Plasma apolipoproteins A (Apo A) and B (Apo B) were also measured used standardized methods. Plasma hs-CRP and IL-6 were determined by the particle-enhanced Immunoturbidime Quantikine ELISA Kit and the Human IL-6 Quantikine HS ELISA Kit, respectively (both from R&D Systems, Inc., Minneapolis, MN, USA).

### 2.4. Statistical Analyses

For a parallel study design, statistical power calculations indicated that 20 participants per group, with an anticipated dropout rate of 5%, were required to detect mean differences of 0.27 ± 0.58 mmol/L (10.4 ± 22.6 mg/dL) in LDL-c (α = 0.05; power = 0.8). The Kolmogorov–Smirnov test showed that smoking, office diastolic BP, and hepatic, muscle, and inflammatory biomarkers did not follow a normal distribution. Accordingly, we used parametric or non-parametric methods as appropriate. Data are expressed as mean and standard deviation (SD) for quantitative normally distributed variables, median and interquartile range for quantitative variables with a skewed distribution, and absolute numbers (percentages) for qualitative variables. Between-group differences in cardiovascular risk factors at baseline were assessed using the chi-square test, ANOVA, or Mann–Whitney U test as appropriate. Repeated-measures ANOVA was used to compare changes in the lipid profile at one and three months by testing the interaction of two factors: Treatment group (nutraceutical combination or placebo) and time with three levels—baseline, one, and three months. The Bonferroni post hoc test was used to adjust for multiple comparisons. We used non-parametric tests for the analyses of changes in safety variables and inflammatory biomarkers and the Mann–Whitney U and Wilcoxon tests to compare skewed continuous variables between the groups and time points, respectively. The level of significance was set at *p* < 0.05. All analyses were performed with SPSS version 22.0 software (IBM Corp., Armonk, NY, USA).

## 3. Results

### 3.1. Characteristics of the Participants

A total of 47 potential candidates were pre-screened for eligibility. Figure 1 shows the study flow diagram. Of these 47 candidates, 5 did not meet the inclusion/exclusion criteria and 2 refused to participate in the trial, therefore 40 participants were randomized into two groups, nutraceutical or placebo. The characteristics of the two groups were well balanced at baseline regarding demographic features, vascular risk factors, and adiposity (Table 1). No participant withdrew before study completion. The mean age was 61.4 years, 67.5% were women and 45% were at moderate cardiovascular risk. Per protocol, all participants had a plasma cholesterol concentration >5.17 mmol/L (200 mg/dL) and LDL-c >2.97 mmol/L (115 mg/dL). One participant was taking oral antidiabetic treatment and 32.5% had treated hypertension. No changes in treatment were reported during the trial. 

### 3.2. Adverse Effects

None of the participants in either group (intervention or placebo) reported any relevant adverse effects and none dropped out from the study. Table 2 shows the baseline values and three-month changes of safety laboratory analyses, including liver, renal, and muscle function tests. No statistically significant differences were observed between the changes after the nutraceutical combination and those after the placebo.

### 3.3. Changes in the Lipid Profile and Other Cardiovascular Risk Factors

Table 3 and Figure 2 show the changes in the lipid profile along the study. Significant reductions in total cholesterol, LDL-c, non-HDL-c, and apolipoprotein B (Apo B) concentrations were observed at one and three months in the nutraceutical group compared to the placebo group (*p*_int_ < 0.001; all). Compared to baseline, total plasma cholesterol decreased by −11.4% and −14.1% at one and three months in the nutraceutical group, respectively, while these values decreased by −3.1% in both measurements in the placebo group. The percent reduction of LDL-c was higher at one and three months in the nutraceutical group (19.8% and 19.7%) compared to the placebo group (4.4% and 6.7%). Apo B changes paralleled those of LDL-c, decreasing in the nutraceutical group by 12.4% and 13.5% at one and three months, respectively, compared with a 2.9% reduction in the placebo group at both time points (*p* < 0.001). No significant inter- and intra-group differences were observed in plasma triglycerides, HDL-c, or Apo A1 (*p* > 0.05; all).

### 3.4. Changes in Serum Inflammatory Markers

Table 4 shows the changes in hs-CRP and IL-6 in the two study groups. Plasma hs-CRP concentrations significantly decreased in the nutraceutical group (*p* = 0.021), but not in the placebo group. Plasma IL-6 concentrations did not change significantly in either group. 

## 4. Discussion

According to recent guidelines, patients with mild-to-moderate hypercholesterolemia at low-to-moderate cardiovascular risk who already follow a healthy diet and perform regular physical activity may be treated with cholesterol-lowering functional foods or nutraceuticals to help achieve target LDL-c levels [3]. In the present randomized clinical trial, we demonstrate that a novel nutraceutical combination including phytosterols, RYR, hydroxytyrosol from olive leaf extract (*O. europaea*), and vitamin E reduces total cholesterol, LDL-c, and Apo B concentrations 14–20% after intervention for three months. The changes observed in LDL-c were in the range of those obtained with other recognized cholesterol-lowering nutraceuticals. Thus, recent reviews of functional foods and nutraceuticals targeting plasma cholesterol concluded that those currently available, either alone or in combination, effectively reduced LDL-c by 5–25% [7,8,9,10,11,12].

Phytosterols (1.5 g) are one component of Aquilea Colesterol^®^. Among functional foods with a cholesterol-lowering action, those enriched with phytosterols have been the topic of many experimental and clinical investigations that have substantiated the safety, efficacy, and cost effectiveness of these agents [21]. In a meta-analysis of 124 trials, doses of phytosterols from 0·6 to 3·3 g/day reduced LDL-c between 6% and 12% [22]. Phytosterols may also reduce plasma triglycerides when they are elevated, but have no effect on HDL-c [23]. These compounds are structurally related to cholesterol, although they have more hydrophobic and bulkier molecules, which bestow on them a higher affinity for intestinal micelles than that of cholesterol. As a consequence, cholesterol is displaced from the micelles and the amount available for absorption by the Niemann–Pick C1-like 1 (NPC1L1) transporter is limited [21]. Phytosterols may interfere with the absorption of carotenoids, with attendant small reductions in plasma levels, but increasing consumption of fruits and vegetables suffices to restore circulating carotenoid concentrations [24]. These compounds have demonstrated anti-inflammatory effects in in vitro studies and in experimental animal models, but the results of clinical studies assessing the response of inflammatory markers such as CRP and cytokines to phytosterol intake have been inconsistent [25,26,27,28]. In our study, CRP, but not IL-6, was significantly reduced in the nutraceutical arm compared to the placebo (Table 4). CRP is manifestly the key inflammatory biomarker examined when assessing the relationship between systemic inflammation and CVD and is considered a predictor of future cardiovascular events [29]. The beneficial effect of Aquilea Colesterol^®^ on CRP can be attributed in part to phytosterols, but other components of the nutraceutical combination, such as RYR, hydroxytyrosol, and vitamin E, might have played a role.

Another component of the tested nutraceutical is RYR, a rice fermentation product of the fungus *Monascus purpureus*, used as a dietary supplement and herbal remedy in China for centuries [30]. Its active ingredient is monacolin K, which was the first statin drug (lovastatin) isolated and approved for treatment of hypercholesterolemia. As such, its mode of action is inhibition of cholesterol synthesis via competitive inhibition of 3-hidroxi-3-metil-glutaril-CoA (HMG-CoA) reductase, with ensuing reduction of total cholesterol and LDL-c. [31]. A meta-analysis of 13 randomized, placebo-controlled clinical trials including a total of 804 dyslipidemic individuals consuming RYR for periods ranging from 4 weeks to 12 months revealed significant mean reductions of 0.97 (95% CI, 1.13 to 0.80) mmol/L in total cholesterol and of 0.87 (95% CI, −1.03 to 0.71) mmol/L in LDL-c [32]. However, there was great heterogeneity among studies, RYR was given alone or in combination with other nutraceuticals, and the doses of RYR varied 18-fold, hence the conclusions of this meta-analysis must be taken with caution. Additionally, while these trials have shown RYR to be safe and well tolerated, most studies were small and of short duration. In that meta-analysis, no relevant side effects were reported at the various RYR doses administered. In fact, compared to placebo, RYR was not associated with any relevant changes of plasma creatine kinase (CK) or glucose concentrations [33]. The monacolin K content in the nutraceutical tested in our trial for daily administration was 10 mg, which is equivalent to the same dose of lovastatin. Even though there have been occasional reports of muscle pain in patients taking RYR, a recent meta-analysis of RYR trials using different doses in patients taking or not statins concluded that this treatment is safe and unassociated with increased musculoskeletal adverse effects [34]. RYR should not be consumed with grapes, since the blood concentrations of monacolin K may increase 5–20 fold, caution must be exercised in patients with renal or liver failure, and should be avoided by pregnant women and children [34].

Aquilea Colesterol^®^ also contains hydroxytyrosol, a polyphenol abundant in extra virgin olive oil. Randomized feeding trials have concluded that the Mediterranean diet supplemented with extra virgin olive oil significantly reduces plasma CRP and IL-6 concentrations after intervention for three months [35] and one year [36]. Hydroxytyrosol has also demonstrated anti-inflammatory effects in mouse models [37]. Thus, the presence of hydroxytyrosol in the nutraceutical combination used in the present study may explain in part the observed reduction in hs-CRP. Monacolin K and the vitamin E contained in the nutraceutical may also contribute to this effect [38]. Given that systemic inflammation plays a crucial role in the initiation and progression of atherosclerosis [39], the favorable changes observed in inflammatory markers in our study may represent an anti-atherogenic effect beyond LDL-c reduction.

The overall salutary effect on atherogenic lipid fractions and inflammation observed in our study can be ascribed to the sum of the effects of all the components of this novel nutraceutical. Thus, phytosterols consistently lower total cholesterol and LDL-c in clinical studies [22], but have also shown anti-inflammatory effects in experimental studies [25]; RYR (monacolin K) reduces LDL-c via a statin-like effect and, as such, can also disclose anti-inflammatory actions [31,32,33,38]; and hydroxytyrosol and vitamin E are recognized antioxidant and anti-inflammatory molecules [15,37].

According to a recent meta-analysis, a commercialized nutraceutical combination containing berberine, policosanol, and RYR (200 mg, corresponding to 3 mg monacolin K) decreased total cholesterol by 9.9% and LDL-c by 13.7% [14]. With mean reductions of 13.9% and 19.7%, respectively, the cholesterol-lowering efficacy of the nutraceutical tested in the current trial compares favorably with the results of this meta-analysis, probably because of a higher dose of monacolin K in Aquilea Colesterol^®^ compared with that nutraceutical. A 15.6% LDL-c reduction was observed in a recent clinical trial of a novel nutraceutical also containing 3 mg monacolin K plus various antioxidants [40]. In this trial hs-CRP was not lowered, but LDL oxidation was reduced and endothelial function improved, pointing to an added cardiovascular benefit of such nutraceutical combinations.

## 5. Conclusions

In summary, the results of the present study demonstrate cholesterol-lowering and anti-inflammatory effects of a nutraceutical combination of phytosterols, RYR, hydroxytyrosol, and vitamin E. Further studies with a longer duration of intervention are warranted. In the meantime, the increasing body of knowledge on nutraceuticals such as Aquilea Colesterol^®^ indicate that these products may be a useful adjunct to a healthy diet in the management of mild-to-moderate hypercholesterolemia in individuals at low-to-moderate cardiovascular risk.

## Figures and Tables

**Figure 1 nutrients-11-00949-f001:**
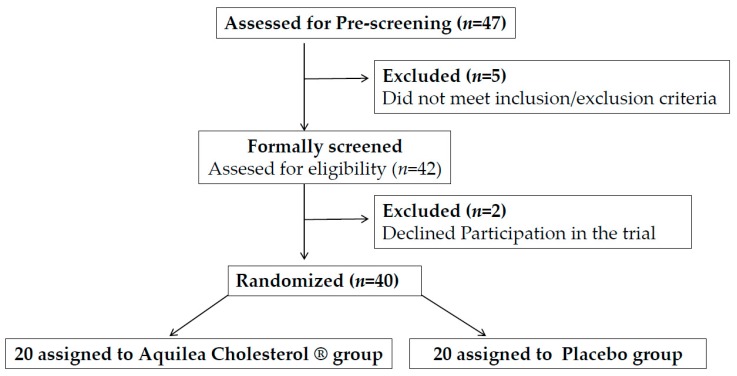
Flowchart of participants.

**Figure 2 nutrients-11-00949-f002:**
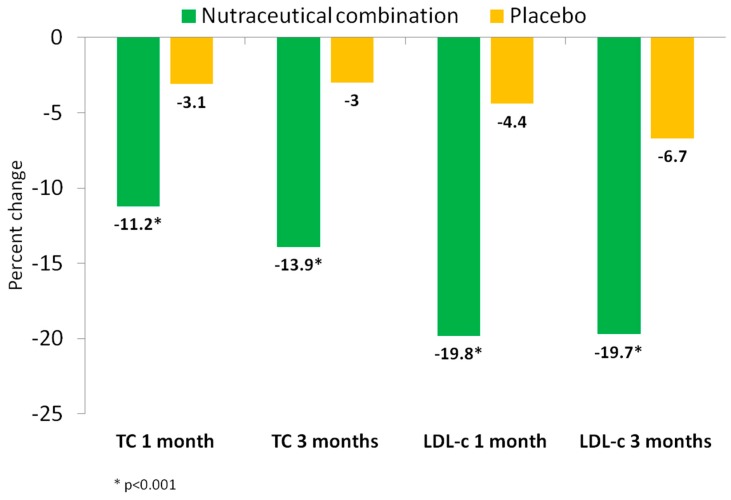
Percent reduction of plasma total cholesterol and LDL-cholesterol by treatment group: Nutraceutical (green), control (yellow). TC: total-cholesterol; LDL: low density lipoprotein.

**Table 1 nutrients-11-00949-t001:** Baseline characteristics of the participants by study group.

	Nutraceutical Combination (*n* = 20)	Placebo (*n* = 20)	*p* *
Age, years	59.3 (9.5)	63.5 (12.3)	0.219
Women, *n* (%)	11 (55)	16 (80)	0.088
Current smokers, *n* (%)	3 (15)	2 (10)	0.500
Weight, kg	74.7 (14.6)	72.0 (17.6)	0.600
Body mass index, kg/m^2^	26.9 (3.8)	27.3 (4.4)	0.159
Waist circumference, cm	97.6 (9.6)	97.1 (10.8)	0.878
Office systolic BP, mm Hg	125.2 (9.9)	127.1 (19.8)	0.562
Office diastolic BP, mm Hg	76.4 (6.7)	73.0 (11.7)	0.275
Total cholesterol, mmol/L	6.08 (0.62)	5.83 (0.47)	0.175
LDL cholesterol, mmol/L	3.91 (0.55)	3.71 (0.48)	0.223
HDL cholesterol, mmol/L	1.44 (0.37)	1.57 (0.14)	0.286
Triglycerides, mmol/L	1.53 (0.87)	1.38 (0.66)	0.271
REGICOR risk scale, *n* (%)			
Low Risk	12 (60)	10 (50)	0.376
Moderate Risk	8 (40)	10 (50)	
MedDiet score	7.6 (2.0)	8.6 (1.9)	0.123

Values are means (SD) except for qualitative variables, which are expressed as *n* (%). BP: blood pressure; LDL: low-density lipoprotein; HDL: high-density lipoprotein; REGICOR: Registre Gironí del Cor; MedDiet: Mediterranean diet. * *p* value for comparisons between groups by chi square test for categorical variables and one-way ANOVA for continuous variables.

**Table 2 nutrients-11-00949-t002:** Baseline and three-month changes in safety laboratory variables by group allocation.

	Nutraceutical Combination (*n* = 20)	Placebo (*n* = 20)	*p* ^†^
Glucose, mmol/L			
Baseline	5.1 (4.7 to 5.5)	5.0 (4.7 to 5.4)	0.314
Change *	−0.28 (−0.26 to 0.37)	0.05 (−0.08 to 0.44)	0.341
Creatinine, μmol/L			
Baseline	0.70 (0.64 to 0.85)	0.71 (0.66 to 0.85)	0.758
Change	0.38 (−3.05 to 5.71)	0.01 (−2.28 to 5.33)	0.327
AST, μkat/L			
Baseline	0.358 (0.337 to 0.395)	0.317 (0.283 to 0.367)	0.096
Change	0.02 (−0.03 to 0.03)	0.01 (−0.05 to 0.07)	0.883
ALT, μkat/L			
Baseline	0.325 (0.235 to 0.445)	0.283 (0.233 to 0.358)	0.414
Change	0.001 (−0.07 to 0.05)	0.001 (−0.04 to 0.08)	0.904
GGT, μkat/L			
Baseline	0.292 (0.250 to 0.608)	0.283 (0.233 to 0.525)	0.863
Change	−0.02 (−0.06 to 0.01)	−0.02 (−0.07 to 0.05)	0.799
LDH, μkat/L			
Baseline	2.867 (2.572 to 3.283)	2.950 (2.683 to 3.383)	0.863
Change	0.03 (−0.08 to 0.20)	−0.02 (−0.26 to 0.21)	0.311
CK, μkat/L			
Baseline	1.600 (1.353 to 1.958)	1.383 (1.058 to 1.700)	0.157
Change	0.06 (−0.17 to 0.23)	−0.13 (−0.41 to 0.16)	0.253
Aldolase, μkat/L			
Baseline	0.063 (0.045 to 0.080)	0.057 (0.048 to 0.072)	0.708
Change	−0.01 (−0.02 to 0.02)	0.01 (−0.02 to 0.02)	0.568

Values are medians (interquartile range). AST: Aspartate aminotransferase; ALT: Alanine aminotransferase; GGT: Gamma glutamyl transpeptidase; LDH: Lactate dehydrogenase; CK: Creatine kinase. * *p* value for comparisons between baseline and three months with the Wilcoxon T test. ^†^
*p* value for comparisons between groups with the Mann–Whitney U test for continuous variables.

**Table 3 nutrients-11-00949-t003:** Baseline and final values of lipids and apolipoproteins by group allocation.

	Baseline	One Month	Three Months	*p* Group	*p* Time	*p* Interaction
Total cholesterol, mmol/L						
Nutraceutical	6.07 (5.83 to 6.32) ^a^	5.38 (5.12 to 5.64) ^b^	5.22 (5.58 to 6.08) ^b^	0.326	<0.001	<0.001
Placebo	5.83 (5.58 to 6.08)	5.66 (5.40 to 5.92)	5.66 (5.38 to 5.94)			
LDL-cholesterol, mmol/L						
Nutraceutical	3.90 (3.68 to 4.14) ^a^	3.32 (3.10 to 3.54) ^b^	3.13 (2.92 to 3.35) ^b^	0.192	<0.001	<0.001
Placebo	3.71 (3.47 to 3.94)	3.56 (3.34 to 3.78)	3.59 (3.38 to 3.78)			
HDL-cholesterol, mmol/L						
Nutraceutical	1.44 (1.26 to 1.62)	1.47 (1.30 to 1.63)	1.49 (1.31 to 1.66)	0.432	0.933	0.337
Placebo	1.57 (1.40 to 1.75)	1.55 (1.39 to 1.72)	1.54 (1.37 to 1.72)			
Triglycerides, mmol/L						
Nutraceutical	1.53 (1.19 to 1.86)	1.29 (1.08 to 1.50)	1.35 (1.11 to 1.60)	0.251	0.058	0.507
Placebo	1.26 (0.92 to 1.60)	1.19 (0.97 to 1.39)	1.13 (0.89 to 1.38)			
Non-HDL-cholesterol, mmol/L						
Nutraceutical	4.63 (4.37 to 4.90) ^a^	3.91 (3.63 to 4.19) ^b^	3.73 (3.47 to 3.99) ^b^	0.691	<0.001	<0.001
Placebo	4.26 (3.99 to 4.53)	4.10 (3.83 to 4.38)	4.11 (3.85 to 4.38)			
Apo B, mg/dL						
Nutraceutical	116.8 (110.4 to 123.2) ^a^	102.3 (94.8 to 109.8) ^b^	101.0 (94.6 to 107.4) ^b^	0.909	<0.001	<0.001
Placebo	108.3 (101.9 to 114.7)	105.1 (97.6 to 112.7)	105.1 (98.7 to 111.5)			
Apo A1, mg/dL						
Nutraceutical	154.4 (144.0 to 164.7)	160.3 (149.9 to 170.6)	160.3 (149.5 to 171.0)	0.251	0.226	0.338
Placebo	157.1 (146.8 to 167.5)	157.8 (147.5 to 168.2)	157.4 (146.6 to 168.1)			

Values are means (SD). LDL: low-density lipoprotein; HDL: high-density lipoprotein; Apo B: apolipoprotein B; Apo A1: apolipoprotein A1. *p* value for one-way repeated-measures ANOVA for continuous variables (Group = treatment allocation; Time = baseline, one month, three months), ^a,b^ Different superscript letters in the same row mean that values were significantly different, *p* < 0.05, by ANOVA simple effect analyses with Bonferroni correction.

**Table 4 nutrients-11-00949-t004:** Baseline levels and changes in anti-inflammatory parameter variables in the nutraceutical and placebo groups.

	Nutraceutical Combination (*n* = 20)	Placebo (*n* = 20)	*p* *
hs-CRP, mg/dL			
Baseline	0.16 (0.04 to 0.45)	0.13 (0.05 to 0.20)	0.925
Change	−0.01 (−0.14 to 0.03)	0.04 (−0.02 to 0.16)	0.021
Interleukin-6, pg/mL			
Baseline	1.65 (2.54 to 3.89)	2.01 (2.65 to 3.82)	0.512
Change	−0.06 (−0.57 to 0.70)	0.09 (−0.57 to 0.34)	0.857

Values are medians (interquartile ranges). hs-CRP: high-sensitive C-reactive protein. * *p* value for comparisons between groups by Mann–Whitney U test.

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
