# Peer review of "Effects of a Novel Nutraceutical Combination (Aquilea Colesterol®) on the Lipid Profile and Inflammatory Biomarkers: A Randomized Control Trial"

_nutrients, 2019, doi:10.3390/nu11050949_

Reviewer 1 Report

Title: Effects of a novel nutraceutical combination (Aquilea 2 Colesterol®) on lipid profile and inflammatory 3 biomarkers: A randomized control trial.

Comment to the Authors:

The manuscript by Domenech does not contain enough high quality data.

General consideration:

The authors should contact an editing agency before re-submission, because the manuscript needs a revision by an English-speaking editor. 

Author Response

REVIEWER 1 

Title: Effects of a novel nutraceutical combination (Aquilea 2 Colesterol®) on lipid profile and inflammatory 3 biomarkers: A randomized control trial.

Comment to the Authors:

The manuscript by Domenech does not contain enough high quality data.

We have deeply reviewed the initial version of our manuscript to better explain the extension of the changes observed in the lipid profile after the 3-month intervention with the nutraceutical combination. As stated in the manuscript, the European Society of Atherosclerosis and the European Society of Cardiology guidelines for the clinical management of dyslipidemia state that low-to-moderate risk subjects can benefit from supplementation with functional foods or nutraceuticals. Among the nutraceuticals with demonstrated effects on hyperlipidemia, the most commonly studied are plant sterols or phytosterols, red yeast rice (RYR), omega-3 fatty acids, soluble fibers, berberine, and policosanol, although only the first four compounds have been recognized by the Food and Drug Administration (FDA) and the European Food Safety Authority (EFSA) as safe and effective in cholesterol lowering. The EFSA panel also considers that 5 mg of hydroxytyrosol and its derivatives (e.g., oleuropein and tyrosol) from olive oil may contribute to normalizing high-density lipoprotein cholesterol (HDL-c) concentrations and protecting against LDL oxidation. Thus, nutraceutical combination such as Aquilea Colesterol may be used to improve the lipid profile in individuals at low-to-moderate cardiovascular risk who maintain plasma cholesterol levels >5.17 mmol/L (>200 mg/dl) and/or LDL-c >2.97 mmol/L (>115 mg/dl) despite a healthy diet and overall lifestyle. In these individuals, pharmacological treatments (i.e., statins) are not indicated and nutraceuticals can fill the gap and help reduce cardiovascular risk. Not many nutraceutical combinations have been tested in randomized clinical trials.

General consideration:

The authors should contact an editing agency before re-submission, because the manuscript needs a revision by an English-speaking editor. 

We apologize for the version submitted. The manuscript has been thoroughly revised by a clinical expert in Lipidology in order to increase the scientific value of the message and also by a native English speaker to improve the quality of the writing. 

Reviewer 2 Report

The manuscript entitled ‘Effects of a novel nutraceutical combination (Aquilea Colesterol®) on lipid profile and inflammatory biomarkers: A randomized control trial’ by Domenech et al., reports the results of a trial on lipid and inflammatory markers in patients with low (<5%) or moderate (5-9%) coronary risk. The current study is interesting as it is reports the effects of Aquilea Colesterol on toxicity, changes in lipid profile and inflammatory biomarkers.

General comments

1.    The diminution of inflammatory biomarkers is questionable. The authors used two biomarkers – hsCRP and IL6. hsCRP was downregulated in patients treated with nutraceutical, but not IL6. And no other inflammatory biomarkers have been tested. This should be discussed better with few more references in the literature.

2.    There are many punctuation and grammatical errors. The paper would be significantly clearer if the authors seek some professional editing help from someone with full professional proficiency in English. 

Specific comments

1.    Line 44, replace > 25mg/m2 with >25mg/m2

2.    The abbreviation LDL-c should be consistently used after defining it first time.

3.    Please add one or two sentences to describe nutraceuticals in the introduction.

4.    Line 95, replace lefaf with leaf.

5.    Independent search by the reviewer for the ISRCTN0163481 did not find the trial. Is this the correct number?

6.    In the line 139, the degree symbol in -80°C is written incorrectly.

7.    The formula of the LDL-c should be mentioned with some references of the previous usage of this formula.

8.    Figure 2, the graph is pasted improperly. Please follow journal instructions.

9.    Due to the lack of changes in IL6, the Table 4 could be more informative if shown as bar graph / box – whisker / dot plot.

10. Line 265, please mention the amount of the doses administered.

11. Extended discussion on the comparison of the doses of individual components in Aquilea Colesterol vs effects of individual components (not in combination) and also some reasoning about doing long term studies will be helpful.

Author Response

REVIEWER 2

The manuscript entitled ‘Effects of a novel nutraceutical combination (Aquilea Colesterol®) on lipid profile and inflammatory biomarkers: A randomized control trial’ by Domenech et al., reports the results of a trial on lipid and inflammatory markers in patients with low (<5%) or moderate (5-9%) coronary risk. The current study is interesting as it is reports the effects of Aquilea Colesterol on toxicity, changes in lipid profile and inflammatory biomarkers.

Thank you for your time and comments. Please find enclosed our answers under each question or comment.

General comments

1.    The diminution of inflammatory biomarkers is questionable. The authors used two biomarkers – hsCRP and IL6. hsCRP was downregulated in patients treated with nutraceutical, but not IL6. And no other inflammatory biomarkers have been tested. This should be discussed better with few more references in the literature.

As stated, hs-CRP and IL-6 are the two inflammatory biomarkers commonly used in clinical trials and hs-CRP in epidemiological studies. Both decreased after the intervention, but the differences only attained statistical significance for hs-CRP. These aspects are better discussed in the revised version (lines 284-291) and 3 more references have been added (see references 32, 33, 34, 35, 36) 

2.    There are many punctuation and grammatical errors. The paper would be significantly clearer if the authors seek some professional editing help from someone with full professional proficiency in English. 

We apologize for the errors. The new version has been extensively revised by an expert lipidologist and by a native English speaker.

Specific comments

1.    Line 44, replace > 25mg/m2 with >25mg/m2

The word has been replaced. Line 44.

2.    The abbreviation LDL-c should be consistently used after defining it first time.

Done.

3.    Please add one or two sentences to describe nutraceuticals in the introduction.

We have added the following sentence in the introduction: “The cholesterol-lowering nutraceuticals are compounds derived from foods or natural molecules that target metabolic pathways leading to a reduction in body cholesterol pools and are commercialized in in pills, powder or other medical forms and sometimes are consumed in fortified foods”. Lines 62-63.

4.    Line 95, replace lefaf with leaf.

Done. Now line 93.

5.    Independent search by the reviewer for the ISRCTN0163481 did not find the trial. Is this the correct number?

Excuse the typo. The correct number of the ISRCTN register is 15149502 (line 989.

6.    In the line 139, the degree symbol in -80°C is written incorrectly.

Corrected. Now in line 136.

7.    The formula of the LDL-c should be mentioned with some references of the previous usage of this formula.

We have added the formula to estimate LDL-c and included the corresponding reference.

8.    Figure 2, the graph is pasted improperly. Please follow journal instructions.

We have corrected revised the Figure and pasted it properly.

9.    Due to the lack of changes in IL6, the Table 4 could be more informative if shown as bar graph / box – whisker / dot plot.

Since the changes are small (although significant) we prefer to keep Table 4 as in the first submission

10. Line 265, please mention the amount of the doses administered.

The following sentences has been added in the discussion: “However, there was great heterogeneity among studies, RYR was given alone or in combination with other nutraceuticals, and the doses of RYR varied 18-fold, hence the conclusions of this meta-analysis must be taken with caution”. See lines 270-273 in the new version.

11. Extended discussion on the comparison of the doses of individual components in Aquilea Colesterol vs effects of individual components (not in combination) and also some reasoning about doing long term studies will be helpful.

Thank you for your suggestion. Now we have included in the discussion, the comparison of the doses used of each component of the nutraceutical analyzed with the effects observed in other studies and also a comparison of the effects of Aquilea Colesterol on lipids with the results on a meta-analysis (ref. 11) of another nutraceutical combination including berberine, policosanol and RYR (lines 294-299).

Reviewer 3 Report

The author designed a randomized, double-blind study to assess the effect of a novel nutraceutical combination on the total cholesterol and LDL-c levels in patients with primary hypercholesterolemia with low-to-moderate cardiovascular risk.  The data at baseline, at 1 and 3 months post nutraceutical/placebo intervention were analyzed. The novel nutraceutical intervention decreased total cholesterol, LDL-c, and apolipoprotein B compared to the placebo group. However, it is not very clear that what the better effects of this novel nutraceutical combination are on treating hypercholesterolemia in patients with low-to-moderate cardiovascular risk compared to previously reported nutraceutical treatment (combination or single component).

Specific comments:

1. Can you please explain why plasma creatine kinase (CK) concentration is significantly higher in the nutraceutical group compared to placebo group (p=0.007)? Does this means that the nutraceutical intervention has side effect?

2. It is confused about p group, p time and p interaction in Table 3. Can you explain these p value? For example, what is the meaning if p group =0.326 for total cholesterol? Does it mean that total cholesterol of nutraceutical group at baseline, 1 month and 3 months was not significantly different from placebo group? Is p Time (<0.001) significant for nutraceutical group or placebo group?

3. Can you please explain the meaning of changes of hs-CRP and IL-6 in Table 4? Do you compare hs-CRP at baseline vs. 3 months post nutraceutical intervention? Can you please explain the meaning of p value in Table 4? For example, what is the meaning of p=0.925?

Author Response

REVIEWER 3

The author designed a randomized, double-blind study to assess the effect of a novel nutraceutical combination on the total cholesterol and LDL-c levels in patients with primary hypercholesterolemia with low-to-moderate cardiovascular risk.  The data at baseline, at 1 and 3 months post nutraceutical/placebo intervention were analyzed. The novel nutraceutical intervention decreased total cholesterol, LDL-c, and apolipoprotein B compared to the placebo group. However, it is not very clear that what the better effects of this novel nutraceutical combination are on treating hypercholesterolemia in patients with low-to-moderate cardiovascular risk compared to previously reported nutraceutical treatment (combination or single component).

Thank for your comment. Following your suggestion we have included now in the discussion, the comparison of the doses used of each component of the nutraceutical tested with the effects observed in other studies and also a comparison of the effects of Aquilea Colesterol on lipids with the results of a meta-analysis (ref. 11) of another nutraceutical combination including berberine, policosanol and RYR (lines 294-299).

Specific comments:

1. Can you please explain why plasma creatine kinase (CK) concentration is significantly higher in the nutraceutical group compared to placebo group (p=0.007)? Does this means that the nutraceutical intervention has side effect?

Thank you for this comment. The higher 3-month CK value in the nutraceutical group compared with the placebo group was due to higher baseline levels in the former. Now we show changes (which obviously account for baseline levels) rather than 3-month values and, as seen in revised Table 2, there are no statistically significant differences.

2. It is confused about p group, p time and p interaction in Table 3. Can you explain these p value? For example, what is the meaning if p group =0.326 for total cholesterol? Does it mean that total cholesterol of nutraceutical group at baseline, 1 month and 3 months was not significantly different from placebo group? Is p Time (<0.001) significant for nutraceutical group or placebo group?

We realize that the results of the ANOVA tests may be somewhat confusing. In this statistical analysis we obtain “3 p values”, one is related to changes in each group, another refers to changes between baseline and final with the two groups taken together, and the “P interaction” is the critical one since it refers to the comparison of changes between the two groups. In the current trial, the P interaction < 0.001 indicates that the reduction in plasma lipid concentrations (total cholesterol and LDL-c) observed after the intervention with the nutraceutical combination was significantly different from the changes observed in the placebo group.    

3. Can you please explain the meaning of changes of hs-CRP and IL-6 in Table 4? Do you compare hs-CRP at baseline vs. 3 months post nutraceutical intervention? Can you please explain the meaning of p value in Table 4? For example, what is the meaning of p=0.925?

In Table 4, in the first row we have included the baseline values of plasma hs-CRP between both groups (intervention and placebo). The p value of 0.925 denotes that there were no differences in the baseline concentration of this inflammatory biomarker between the nutraceutical and the placebo groups. However, after 3 months of intervention, we observed a significant reduction in plasma hs-CRP concentration with the nutraceutical combination, compared to the changes observed in the placebo group (p=0.021). On the other hand, there were neither baselines nor post-treatment differences between groups in Interleukin-6.

Round  2

Reviewer 1 Report

I respect the efforts made by Domenech and co-workers to improve the manuscript, but still my concern about the fact that it does not contain enough high quality data.

Author Response

We would like to thank the reviewer for the time spent reviewing again our manuscript. In the new version we have revised again the English language and we have modified extensively the introduction, which now is more focused, and in part the discussion, to upgrade the scientific value of the manuscript.  

We, indeed, believe that the results of our study with a novel nutraceutical combination are valid and contribute to the literature in the field, providing another therapeutic approach for individuals with moderate hypercholesterolemia at low-to-moderate cardiovascular risk in whom treatment with statins is not indicated or who do not tolerate statins. When reviewing the literature available on randomized placebo controlled trials with cholesterol-lowering nutraceutical combinations, we have found several position papers (Nutr Rev. 2017; 75: 731-767; Pharmacological Res 2018;134:51-60; JACC 2018;72:96-118), reinforcing the usefulness of such combinations in the treatment of primary dyslipidemia or statin intolerant patients. These papers are cited now in our manuscript. The results we obtained with the novel nutraceutical combination used are somewhat better than other combinations concerning LDL-c and CRP responses, thus we believe that this fact may interest colleagues caring for patients at cardiovascular risk and the paper merits to be published in a journal such as Nutrients.

Reviewer 3 Report

I am appreciated that the author made the changes according to the suggestions. I have no more comments.

Author Response

We would like to thank the reviewer for his/her efforts to improve the quality of the manuscript and the final comment.